# A New Polycaprolactone-Based Biomembrane Functionalized with BMP-2 and Stem Cells Improves Maxillary Bone Regeneration

**DOI:** 10.3390/nano10091774

**Published:** 2020-09-08

**Authors:** Céline Stutz, Marion Strub, François Clauss, Olivier Huck, Georg Schulz, Hervé Gegout, Nadia Benkirane-Jessel, Fabien Bornert, Sabine Kuchler-Bopp

**Affiliations:** 1INSERM (French National Institute of Health and Medical Research), UMR 1260, Regenerative NanoMedicine (RNM), FMTS, 67000 Strasbourg, France; celine.stutz@etu.unistra.fr (C.S.); marion.strub@chru-strasbourg.fr (M.S.); francois.clauss@chru-strasbourg.fr (F.C.); o.huck@unistra.fr (O.H.); herve.gegout@unistra.fr (H.G.); nadia.jessel@inserm.fr (N.B.-J.); fabien.bornert@unistra.fr (F.B.); 2Faculté de Chirurgie Dentaire, Université de Strasbourg (UDS), 8 rue Ste Elisabeth, 67000 Strasbourg, France; 3Pôle de Médecine et Chirurgie Bucco-Dentaires, Pediatric Dentistry, Hôpitaux Universitaires de Strasbourg (HUS), 1 place de l’Hôpital, 67000 Strasbourg, France; 4Pôle de Médecine et Chirurgie Bucco-Dentaires, Periodontology, Hôpitaux Universitaires de Strasbourg (HUS), 1 place de l’Hôpital, 67000 Strasbourg, France; 5Core Facility Micro- and Nanotomography, Biomaterials Science Center (BMC), Department of Biomedical Engineering, University of Basel, Gewerbestrasse 14, 4123 Allschwil, Switzerland; georg.schulz@unibas.ch; 6Pôle de Médecine et Chirurgie Bucco-Dentaires, Oral Medicine and Oral Surgery, Hôpitaux Universitaires de Strasbourg (HUS), 1 place de l’Hôpital, 67000 Strasbourg, France

**Keywords:** biomembrane, bone regeneration, nanoreservoirs, smart implant, stem cells

## Abstract

Oral diseases have an impact on the general condition and quality of life of patients. After a dento-alveolar trauma, a tooth extraction, or, in the case of some genetic skeletal diseases, a maxillary bone defect, can be observed, leading to the impossibility of placing a dental implant for the restoration of masticatory function. Recently, bone neoformation was demonstrated after in vivo implantation of polycaprolactone (PCL) biomembranes functionalized with bone morphogenic protein 2 (BMP-2) and ibuprofen in a mouse maxillary bone lesion. In the present study, human bone marrow derived mesenchymal stem cells (hBM-MSCs) were added on BMP-2 functionalized PCL biomembranes and implanted in a maxillary bone lesion. Viability of hBM-MSCs on the biomembranes has been observed using the “LIVE/DEAD” viability test and scanning electron microscopy (SEM). Maxillary bone regeneration was observed for periods ranging from 90 to 150 days after implantation. Various imaging methods (histology, micro-CT) have demonstrated bone remodeling and filling of the lesion by neoformed bone tissue. The presence of mesenchymal stem cells and BMP-2 allows the acceleration of the bone remodeling process. These results are encouraging for the effectiveness and the clinical use of this new technology combining growth factors and mesenchymal stem cells derived from bone marrow in a bioresorbable membrane.

## 1. Introduction

Several events, such as tooth extraction, and diseases such as periodontitis induce significant alveolar bone loss [1,2]. Such bone loss could also be induced by life-threatening diseases such as cancers or rare bone structural or metabolic diseases characterized by low bone mass or abnormal and excessive bone remodeling [3]. Management of such bone defects is of tremendous importance as bone volume and osseous tridimensional configuration influence treatment choice and its complexity. For instance, dental implant placement requires a remaining alveolar bone of sufficient height and thickness [4]. Reduced quantity or quality of bone will necessitate bone regeneration, bone grafting techniques or a guided bone regeneration approach [5]. Current management of bone defects consists of bone grafting or bone filling with concomitant use of autologous bone and bone substitutes such as particles or blocks of xenogenic or alloiplastic origin [6]. Barrier membranes associated with bone substitutes have a key role in the regeneration of underlying bone defects by bioactive contribution [7]. These barrier membranes must fulfill several requirements, such as biocompatibility, occlusivity, spaciousness, clinical manageability and appropriate integration with the surrounding tissue, but an ideal membrane has not been established yet for clinical applications [8]. Some authors showed that guided bone regeneration can be enhanced by integrating growth factors (GFs) or cells [9,10].

Different strategies have already been suggested. Indeed, (i) the use of growth factors [11]; (ii) the use of post-natal stem cells [12] and (iii) the production of a three-dimensional “scaffold” with controlled architecture [13] have already been evaluated in the context of bone regeneration. Such strategies are based on the self-repair ability of bone throughout life and on osteoinduction and osteoconduction [14]. Stem cell therapy is one of the most promising methods for the bone repair process. It was already demonstrated that mesenchymal stem cells (MSCs) are involved in the biological processes of bone regeneration. Among the multiple types of MSCs, bone marrow derived MSCs (BM-MSCs) display several advantages, including high potential for osteoblastic differentiation [15] and the ability to self-renew [16]. Interestingly, growth factors play an important role in directing MSC differentiation [17].

Bone morphogenic protein-2 (BMP-2) belongs to the TGF-β superfamily. These proteins regulate various biological processes such as organogenesis or tooth and bone development [18]. BMP-2 activates different signaling pathways, like Wnt/β-catenin signaling [19] and the MAPK pathway [20], leading to osteogenesis. BMP-2 is also approved by the Food and Drug Administration for its osteoinductive properties and is currently used to improve bone regeneration outcomes [21].

Polycaprolactone (PCL) is one of the most easily synthesized synthetic polymers to handle and process. It is possible to give it many shapes and to adjust its size very easily due to its viscoelastic properties as well as its low melting temperature [22] and it is commonly used in the context of tissue engineering. PCL is biocompatible and mimics the extracellular matrix. This membrane can be functionalized with active molecules such as drugs or growth factors [23,24,25,26]. To protect these biologically active compounds, nanoreservoir technology has been developed for trapping, protection and stabilization in implantable biomembranes. These nanoreservoirs allow an active and prolonged release of drugs, genes or growth factors following contact between the cells and these nanoreservoirs and allow an active release of the drug [27]. Preliminary studies have demonstrated the efficiency of the implantation of a PCL biomembrane functionalized with BMP-2 and an anti-inflammatory, ibuprofen, in the regeneration of maxillary bone in mice [26]. These first observations revealed a large recruitment of cells on the lesion in the presence of BMP-2 and thus an acceleration of bone regeneration.

The aim of this study was to evaluate the pro-regenerative effects of a biomembrane combining both BMP-2 and hBM-MSCs’ properties in the context of maxillary bone lesion in an attempt to improve its regeneration in a spatially controlled delivery.

## 2. Materials and Methods

### 2.1. Materials

Polycaprolactone (PCL; MW 124 kDa) clinical grade was purchased from Corbion (Corbion, Gorinchem, The Netherlands). BMP-2 (200 ng/mL) was acquired from Euromedex (Euromedex, Souffelweyersheim, France) and chitosan (Protasan UPCL 113, 500 g/mL) from Novamatrix (Novamatrix, Sandvika, Norway).

### 2.2. Preparation of Electrospun Nanofibers and Nanoreservoir Formation

PCL was dissolved in a mixture of dichloromethane/dimethylformamide (DCM/DMF 50/50 *v*/*v*) at 15% wt/v and stirred overnight before use. A standard electrospinning set-up (EC-DIG apparatus, IME Technologies, Eindhoven, The Netherlands) was used to fabricate the PCL scaffolds. The PCL solution was poured into a 5 mL syringe and ejected through a needle with a diameter of 0.5 mm, at a flow rate of 1.2 mL/h, with a programmable pump (Harvard apparatus). A high-voltage power supply (SPELLMAN, SL30P10) was used to set 15 kV at the needle. Aluminum foils (20 × 20 cm^2^), connected to the ground collector at a distance of 17 cm from the needle, were used to collect the electrospun PCL scaffolds, which were dried in a vacuum oven overnight to remove traces of solvent (Figure 1A). The formation of nanoreservoirs on the nanofibers is based on the electrostatic properties of the components. Chitosan (500 μg/mL) and BMP-2 (200 ng/mL) were diluted in MES buffer (40 mM, pH 5.5). All rinse solution used was MES buffer (40 mM, pH 5.5). PCL scaffolds were incubated in the adequate polyelectrolyte or protein solutions for 15 min. Chitosan has a positive charge with a p*K_a_* of 6.5 and BMP-2 solution has a negative charge in the same condition. After each deposition step, the membranes were rinsed for 15 min with the buffer medium. Repeating this protocol ten times allowed the construction of (chitosan/BMP-2)_10_ scaffold (Figure 1B).

### 2.3. Cell Culture

The hBM-MSCs (Promocell, Heidelberg, Germany) ranging from passage 4 to 6 were grown at 37 °C in a humidified atmosphere of 5% CO_2_ in ready-to-use Mesenchymal Stem Cell Growth Medium 2 (Promocell, Heidelberg, Germany). The medium was changed every 2–3 days. When cells reached sub-confluence, they were harvested with trypsin and sub-cultured on cover glasses, on non-functionalized PCL and on BMP-2 functionalized PCL membranes in 24-well plates for 7 days. Before cell seeding, scaffolds were treated with 70% ethanol and sterilized by 30 min exposure to UV light.

### 2.4. Morphological Characterization of Electrospun Nanofibers

SEM allowed us to characterize the morphological structure of the nanofibers, the nanoreservoirs on the PCL biomembrane and the morphology of the hBM-MSCs on the scaffolds after 7 days of culture. Biomembranes were fixed and dehydrated in ethanol baths of increasing concentration (25%, 50%, 75%, 90% and 100%), each for 15 min. They were placed on a specimen holder and fixed with carbon-conductive adhesive tape. Hexamethyldisilazane (HDMS, ThermoFisher Scientific, Illkirch, France) was deposited on the samples. The objective was to observe the nanofibrous substructure, the size and the porosity of the fibers and the distribution and size of the nanoreservoirs using a SEM microscope (Field Emission Scanning Electron Microscope CSEM-FEG INSPECT 50, FEI, Hillsboro, OR, USA).

### 2.5. Viability of Cells

The viability of hBM-MSCs was assessed using a fluorescence-based LIVE/DEAD^®^ assay (LIVE/DEAD^®^ Cell Imaging Kit, Molecular Probes™, Invitrogen, Villebon-sur-Yvette, France). Cells were washed with phosphate-buffered saline (PBS), stained with a solution of calcein AM reagent and EthD-1 reagent mixed in 2 mL of PBS for 10 min, as recommended by the manufacturer. Cells were observed with a microscope (Leica DM4000B) equipped for fluorescence.

### 2.6. Differentiation of the MSCs

#### 2.6.1. Colony Forming Unit Fibroblasts (CFU-f) Test

This test makes it possible to determine the proliferation potential of stem cells by counting the number of cells capable of generating colonies. Variable concentrations of hBM-MSCs (from 40,000 to 50 cells) were seeded in a 6-well plate. The cells were cultured for 14 days in Mesenchymal Stem Cell Growth Medium 2 at 37 °C in a humidified atmosphere of 5% CO_2_. The slides were then rinsed with PBS, fixed with paraformaldehyde (PFA) (4%) for 10 min at 4 °C and stained with hematoxylin-eosin.

#### 2.6.2. Osteocyte Differentiation

Human BM-MSCs were seeded on cover glasses in a 24-well plate at a density of 21,000 cells/cm^2^ and for 3 days in Mesenchymal Stem Cell Growth Medium 2 until confluence. Differentiation was then induced by culturing the cells in osteogenic medium consisting of α Minimum Essential Medium Eagle (MEM), supplemented with glutamine (2 mM), FBS (10%), dexamethasone (10 nM), ascorbic acid (50 μg/mL) and β-glycerophosphate (10 mM). After 21 days of culture, the cells were rinsed with PBS, fixed with ethanol at 70% for 1h at −20 °C and rinsed with distilled water 3 times for 10 min. An Alizarin Red S staining, specific for calcium deposits, was then performed on the coverslips.

#### 2.6.3. Adipocyte Differentiation

Human BM-MSCs were seeded on cover glasses in a 24-well plate at a density of 21,000 cells/cm^2^ and cultured in their proliferation medium until confluence. Then, adipocyte differentiation was induced by 3 cycles of induction/adipocyte maintenance. The cells were first cultured for 3 days in an induction medium composed of a culture medium supplemented with dexamethasone, indomethacin, h-insulin, MCGS (Mensenchymal Cell Growth Supplement), IBMX (3-isobutyl-methyl-xanthine), Ga-1000 (Gentamicin-Amphotericin) and l-glutamine. Then, for the next 3 days, they were cultured in a maintenance medium consisting of a culture medium supplemented with dexamethasone, indomethacin, h-insulin, MCGS and Ga-1000. After the 3 cycles, cells were cultured in the maintenance medium for 7 days, changed every 2–3 days. After PBS rinsing, cells were fixed in PFA (4%) for 10 min at 4 °C. Cells were then rinsed with isopropanol (60%) for 1 min, stained with the red oil staining solution (Oil Red O, 0.5%) for 30 min. A first rinse with isopropanol 60% and 3 rinses with distilled water were performed. Finally, the coverslips were observed under a microscope to detect lipid vesicles present in cells.

#### 2.6.4. Chondrocyte Differentiation

Micromasses of 250,000 hBM-MSCs were formed by centrifugation at 1000× *g* for 5 min. Chondrocyte differentiation was induced by culturing the micromasses for 21 days in a chondrocyte medium consisting of αMEM, supplemented with glutamine (2 mM), SVF (10%), dexamethasone (10 nM), ascorbic acid (50 μg/mL), L-Proline (40 μg/mL) and TGF-β3 (10 ng/mL). After 21 days of culture, the micromasses were fixed with PFA (4%) for 10 min at 4 °C and embedded in Tissue Tek^®^ (Sakura, Villeneuve d’Ascq, France). Cryostat sections were stained with Alcian Blue to visualize the acidic glycosaminoglycans (GAGs) specifically present in the cartilage.

### 2.7. In Vivo Micro-Surgical Protocol

The experimental protocol fulfilled the authorization of the “Ministère de l’Enseignement Supérieur et de la Recherche” under the agreement number 01716.02. The Ethics Committee of Strasbourg, named “Comité Régional d’Ethique en Matière d’Expérimentation Animale de Strasbourg (CREMEAS)”, specifically approved this study. Under general anesthesia, a maxillary bone lesion was created in the diastemal area with a dental bur (500 m) after gingival incision in nude mice. On one side, PCL scaffold functionalized with BMP-2 and supplemented with hBM-MSCs was implanted while, on the other side, PCL supplemented with hBM-MSCs was implanted. The gingiva over the treated bone defect was closed with biological glue (Histoacryl^®^, B. Braun, Rubi, Spain). Three nude mice were used for each implantation time (90, 120 and 150 days) and the collected samples were treated for histology.

### 2.8. X-ray Microtomography

Heads were dissected and fixed overnight in PFA at 4 °C and then immobilized in agar in a 15 mL Falcon tube. X-ray microCT acquisitions were performed using the Nanotom^®^ m system (GE Sensing & Inspection Technologies GmbH, Wunstorf, Germany). Regarding isotropic spatial resolution, the pixel size was 8 μm and the field size approximately 24.6 mm × 19.2 mm. Data acquisition and volume reconstructions were performed using Phoenix Datos × 2.0 software (Phoenix X-ray, GE Sensing & Inspection Technologies GmbH, Wunstorf, Germany).

### 2.9. Histology

Human BM-MSCs cultured for 7 days on PCL biomembranes were fixed for 10 min with 4% paraformaldehyde PFA and stained with hematoxylin-eosin (HE). Maxillaries were fixed for 24 h in 4% PFA, decalcified in Decalcifier II (Leica Microsystems, Nanterre, France) at 37 °C for 2 h under agitation and embedded in paraffin. Serial sections (10 μm) were stained with Gomori trichrome staining and observed on a Leica DM4000B microscope.

## 3. Results

### 3.1. Characterization of the Scaffold

The PCL nanofibrous structure (Figure 1A) and the distribution of BMP-2 nanoreservoirs (Figure 1B) were characterized by scanning electron microscopy (SEM). The PCL scaffolds obtained by electrospinning evidenced a non-woven mesh-like structure with a large surface area per volume ratio (Figure 1A). The fibers were uniform in size and interconnected in order to mimic the natural extracellular matrix. The diameter of fibers was 975 ± 225 nm for the PCL electrospun fibrous membrane. Ten bilayers of (chitosan/BMP-2) were built on the PCL scaffold. Nanoreservoirs were tightly grafted on the surface of the electrospun nanofibers (Figure 1B). The size of the nanoreservoirs indicated by black arrowheads (Figure 1B) has been estimated at 200 ± 40.5 nm.

### 3.2. Characterization of the Mesenchymal Stem Cells

The self-renewal capacity of the stem cells was proven by the colony forming unit test (CFU) (Figure 2A). Colonies appeared as concentric “spots” (Figure 2A, arrows) stained with HE. On the other hand, the multipotency capacity of stem cells derived from human bone marrow was evaluated. After 21 days of culture in an osteogenic differentiation medium, mineralization nodules developed in the cytoplasm of the cells, as observed with Alizarin Red S staining (Figure 2B, arrows). After 21 days of culture in an adipogenic medium, cytoplasmic lipid vacuoles were evidenced by the Oil Red stain (Figure 2C). Finally, after culturing micromasses of hBM-MSCs (Figure 2D) in a chondrogenic medium for 21 days, the acidic glycosaminoglycans (GAGs) of the cartilaginous extracellular matrix were revealed with Alcian Blue (Figure 2D). Altogether, such results confirm the pluripotency of hBM-MSCs.

To evaluate the cytocompatibility of the synthesized PCL biomembranes, hBM-MSCs were cultured on PCL and PCL-BMP-2. Cell morphology and viability after 7 days were observed after HE staining (Figure 3A,B), using LIVE/DEAD staining (Figure 3C,D) and with SEM (Figure 3E,F). In both tested biomembranes, cytocompatibility was very high as only a few dead cells were observed (Figure 3C,D). The SEM analysis showed that the hBM-MSCs were adherent and spread on the surface of PCL fibers with several extensions (Figure 3E,F). These results confirmed the absence of cytotoxicity of the synthesized biomembranes with and without BMP-2.

### 3.3. Maxillary Bone Regeneration

After drilling of a 500 mm diameter defect (Figure 4), osseous lesions were treated on one side with PCL/hBM-MSCs and on the other side with PCL/(BMP-2)_10_/hBM-MSCs. At the same time, controls with unfunctionalized PCL and PCL/(BMP-2)_10_ were carried out to complete the previously obtained results for 30 days of implantation [26], for 90 days of implantation [25] and, shown in Appendix A, for 120 days of implantation.

After 90, 120 and 150 days of implantation, the samples were analyzed by ex vivo micro-CT and histology. In the presence of hBM-MSCs and the presence or not of BMP-2, the lesion was not completely closed after 90 days of implantation (Figure 5). On the other hand, there was more bone remodeling and neoformed bone when the implanted scaffold was functionalized with BMP-2 and hBM-MSCs. In the presence of BMP-2 and hBM-MSCs, the lesion was four times more closed than in the presence of hBM-MSCs alone. The delineation between the native maxillary bone and the neoformed bone matrix was visible and Gomori trichrome staining showed the presence of cells in the bone defect (Figure 5E,F). After 120 days of implantation (Figure 6), the lesion was almost closed on the side where a membrane functionalized with BMP-2 and supplemented with hBM-MSCs was implanted (Figure 6E). On the non-functionalized side with the cells, bone remodeling was shown; however, the lesion was not completely closed. After 120 days, the efficacy of PCL-BMP-2 versus untreated PCL’s efficacy on bone remodeling was also tested. Micro-CT and histology results have shown that the presence of BMP-2 accelerates the regeneration of maxillary bone (Appendix A). Histological staining showed bone remodeling and filling of the bone defect (Figure 6F). After 150 days of implantation, bone regeneration seemed to be more advanced in the presence of a membrane functionalized with BMP-2 and hBM-MSCs; however, bone remodeling was not complete in either case. Indeed, the thickness of the newly formed bone is not equivalent to the thickness of the initial bone (Figure 7).

## 4. Discussion

In this study, we demonstrated the favorable pro-regenerative properties of a PCL biomembrane functionalized with BMP-2 and hBM-MSCs. The placement of such a biomembrane in the bone defect improved significantly the healing rate in comparison with untreated cases and PCL membrane alone, emphasizing its putative clinical interest for the management of maxillary bone defects.

The nanofibrous PCL scaffold, synthesized using the electrospinning method, is an ideal material for bone tissue engineering due to its strength, slow degradation, biocompatibility and biodegradability [28]. It was already established that such biomaterial enhances the expression of the osteoblast phenotype and promotes mineralization; indeed, cells on the nanofibrous scaffold expressed more bone sialoprotein (BSP), which plays a prominent role in the initiation of mineralization [29]. The PCL 3D scaffold shows excellent osseointegration of the membrane into the bone defect [30]. Nevertheless, the functionalization of this biomembrane with the nanoreservoirs allows the encapsulation of a controlled amount of drugs, peptides or other active compounds [27]. This technology is therefore considered interesting to deliver active molecules of therapeutic interest at the pathological site in spatial and time-controlled manners [31]. Regarding the use of BMP-2, Liu et al. [32] demonstrated that a slow release of this growth factor from a scaffold enhances osteoinductivity, highlighting the importance of the rate of release. In our study, the use of PCL that has a slow degradation rate is appropriate to the requested slow release of BMP-2 during bone healing. Nanoreservoirs technique allows also the use of a small and controlled amount of BMP-2. This reduced quantity, as well as the spatial control of the compound delivery, may decrease the risk of side effects that can be induced by a high dose of BMP-2, such as ectopic bone formation, inflammatory complication or tumor formation [21,33].

Other studies have evaluated the value of PCL and BMP-2 in bone regeneration. Kim et al. [34] observed that the release of BMP-2 from their PCL-based scaffold resulted in a remarkable increase in new bone formation. Lee et al. [35] and Yun et al. [36] have shown that the release of BMP-2 leads to an increase in alkaline phosphatase (ALP) activity, ALP being an early osteogenic marker. The release of BMP-2 by their PCL scaffold promotes the differentiation of cells into bone lineage cells. Thus, the association of PCL with BMP-2 has already demonstrated its effectiveness in terms of bone regeneration efficiency.

Mesenchymal stem cells (MSCs) have a self-renewal capacity and can be differentiated into osteogenic, chondrogenic, adipogenic and other lineages. These cells can be isolated and identified from many tissues such as adipose tissue, dermal tissue, intervertebral disc and also bone marrow or dental tissues [37]. The use of MSCs has been an interesting strategy in tissue regeneration for several decades. The differentiation of MSCs is crucial for bone regeneration. Indeed, stem cells respond to local biologic and mechanical environments and differentiate into any cellular component needed [38]. Bone marrow-derived MSCs (BM-MSCs) present many advantages for bone healing: they are particularly easy to isolate, their default pathway results in production in cells of the bone lineage, and there is a lack of ethical controversy associated with their use [39]. Physiologically, chemoattractant molecules released at the bone defect play an essential role in MSC attraction. MSCs are also able to regulate the immune response, which is an asset for bone regeneration. MSCs’ paracrine signals promote angiogenesis by activating MEK/MAPK and PI3K/AKT pathways, leading to changes in the bone tissue microenvironment and thus benefits for osteogenesis [40]. Studies using only unscaffolded MSCs are rare; however, Mashimo et al. [39] suggested that transplantation of BM-MSCs directly in tooth extraction sockets accelerates bone healing.

The differentiation of MSCs can be controlled by the addition of some growth factors such as BMP-2 [41]. BMP-2 induces Runx2 expression, thus promoting osteogenic differentiation [42]. BMP-2 presents interest for the initiation of bone healing [43]; thus, the supply of BMP-2 directly into the bone defect makes it possible to quickly initiate the process of bone regeneration. Different carrier technologies are developed to deliver BMP-2 and MSCs directly in the defect. Different materials are available, such as alginate hydrogel [44], resorbable calcium-based scaffolds [45] or chitosan-alginate composite [46]. The release of BMP-2 directly into the lesion and the delivery of stem cells to the site quickly allows the initiation of the bone regeneration process. Moreover, in addition to allowing localized administration of BMP-2 and MSCs, the PCL scaffold promotes bone healing due to its physical properties discussed above. Stem cell viability on functionalized PCL membranes after 7 days was observed using the LIVE/DEAD test, and cell morphology and cell adhesion were observed at the SEM on PCL and PCL/BMP-2 scaffolds. Functionalization with BMP-2 did not result in any changes in viability and cell adhesion, so PCL membranes functionalized with BMP-2 are non-toxic for the cells. Membranes functionalized only with BMP-2 have already shown their effectiveness in previous studies [25]. BMP-2 plays a key role in the regeneration of the maxillary bone: a remarkable difference is observed when this growth factor is present. The cells are recruited in the lesion and differentiate into cells of the bone line, allowing faster regeneration of the bone tissue [25].

Therefore, the objective of this study was to highlight the possibility of supplementing the implants with human mesenchymal stem cells derived from bone marrow. Both conditions were tested in the same animal in order to be free from inter-individual variability. On one side, a PCL membrane functionalized with BMP-2 supplemented with hBM-MSCs was implanted, and on the other side, a non-functionalized membrane supplemented with hBM-MSCs was implanted for 90, 120 and 150 days. At each time point, the bone defect was further filled by neoformed bone tissue in the presence of stem cells on the membrane functionalized with BMP-2. We found that, 150 days after surgery, the use of a PCL membrane functionalized with BMP-2 and hBM-MSCs showed real effectiveness on bone remodeling and kinetic of bone regeneration at the level of the lesion compared to the non-functionalized membrane with stem cells. The functionalized implant with hBM-MSCs regenerated bone tissue of satisfactory thickness.

The micro-CT allows us to characterize the filling of the lesion at several time points; however, this technique does not bring any information concerning the quality of the neoformed bone tissue. Indeed, the lesion is completely filled 150 days after the surgery; however, additional biomechanical tests should be conducted to ensure that the newly formed tissue is as hard and resistant as the original bone. Bone mineral density could be measured by two-photon X-ray absorptiometry (bone densitometry), for example [47]. In addition, Raman spectroscopy could provide information on the level of hydroxyapatite and collagen in bone tissue [48]. The mouse is the most relevant animal model for this study; indeed, mice are particularly interesting in this field of research because of the possibility to generate genetic mutations, the availability of clinically relevant disease models, short breeding cycles, low costs and fast regeneration [49]. The mouse model used for this project was the nude mouse; indeed, this athymic mouse allows us to implant human cells in a mouse without having immune reactions leading to rejection. The maxillary bone defect is a good model for orofacial bone regeneration. This model can be used in the study of abnormalities related to tooth eruption or in the study of bone regeneration in periodontitis. These results should be completed in order to define the significance of the effectiveness of this device and the molecular mechanisms involved. The design of a scaffold for active release of BMP-2 supplemented with mesenchymal stem cells appears to hold promise in the field of regenerative medicine.

## 5. Conclusions

Bioactive scaffold appears as a promising approach aiming to enhance bone regeneration. The combination of a synthetic PCL scaffold functionalized with BMP-2 and hBM-MSCs allows faster bone regeneration. This sophisticated implant could be a beneficial candidate for maxillary bone regeneration.

## Figures and Tables

**Figure 1 nanomaterials-10-01774-f001:**
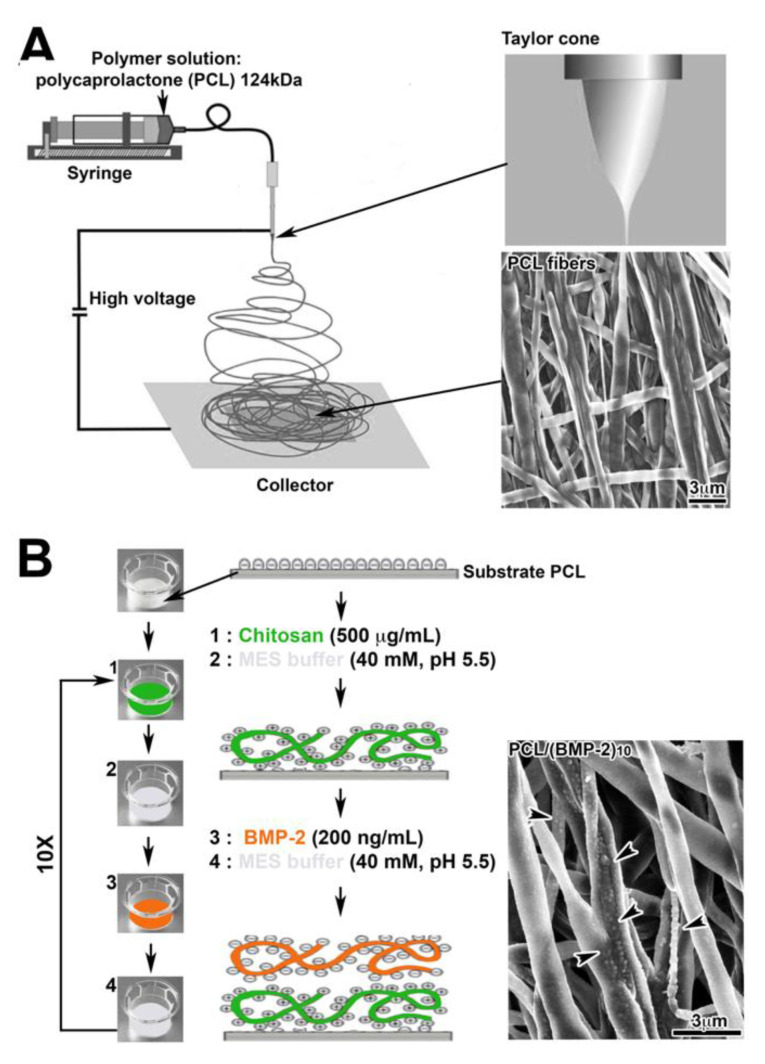
(**A**) Diagram of the electrospinning technique and scanning electron microscopy (SEM) observation of electrospun PCL fibers. (**B**) Diagram of the nanoreservoir functionalization technique and SEM observation of PCL fibers functionalized with BMP-2 (PCL/(chitosan/BMP-2)_10_, 200 ng/mL). Arrowheads indicate the nanoreservoirs containing BMP-2 observed with SEM.

**Figure 2 nanomaterials-10-01774-f002:**
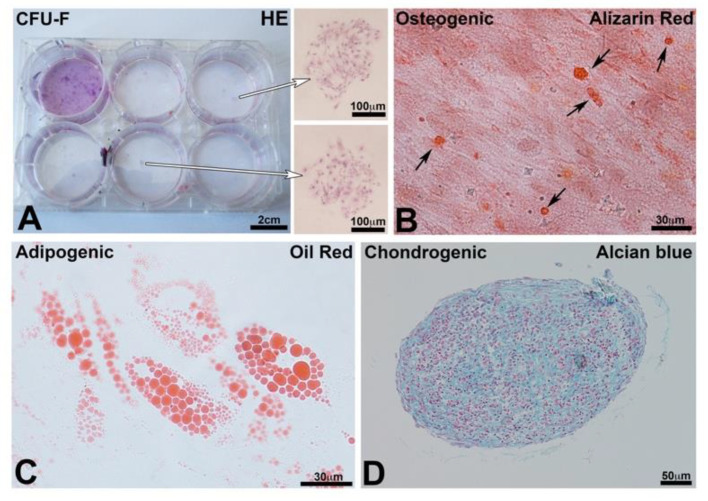
(**A**) Colony forming unit fibroblasts test (CFU-F): staining with HE of CFU formed after 14 days in culture. Arrows indicate 2 colonies. (**B**) Osteogenic differentiation after 21 days in the osteogenic medium; the differentiated cells produced nodules of mineralization stained with Alizarin Red S (arrows). (**C**) Adipocyte differentiation after 21 days in an adipogenic medium. Differentiated cells produced lipid vacuoles stained with Oil Red O. (**D**) Chondrocyte differentiation of the micromasses of hBM-MSCs after 21 days. Alcian Blue visualized the glycosaminoglycans (GAGs) specifically present in the cartilage.

**Figure 3 nanomaterials-10-01774-f003:**
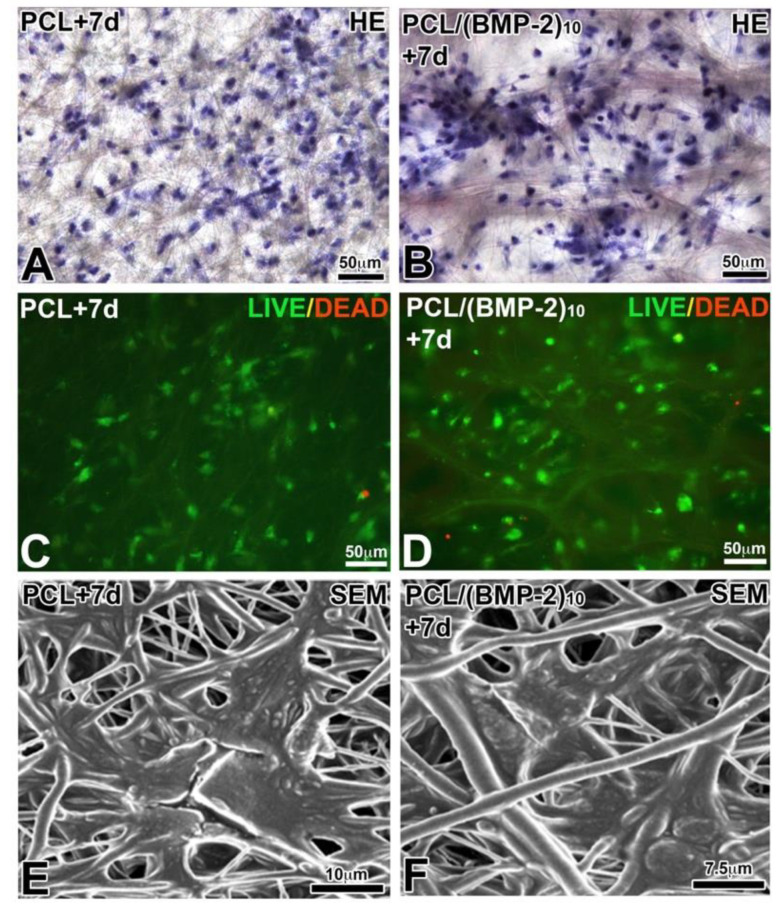
Biocompatibility of PCL and bioactive scaffold for hBM-MSCs after 7 days of culture. HE staining (**A**,**B**), LIVE/DEAD viability test (**C**,**D**) and SEM observation (**E**,**F**). PCL and BMP-2 were not toxic to hBM-MSCs.

**Figure 4 nanomaterials-10-01774-f004:**
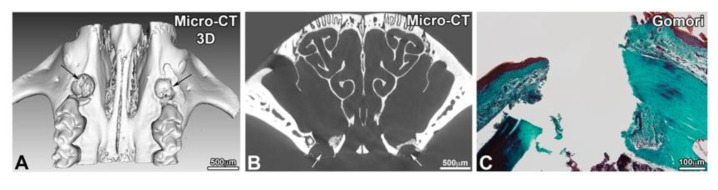
Observation of maxillary bone lesion after 3D reconstruction (**A**) of X-ray microtomography images (**B**) and after histological staining with Gomori trichrome staining (**C**).

**Figure 5 nanomaterials-10-01774-f005:**
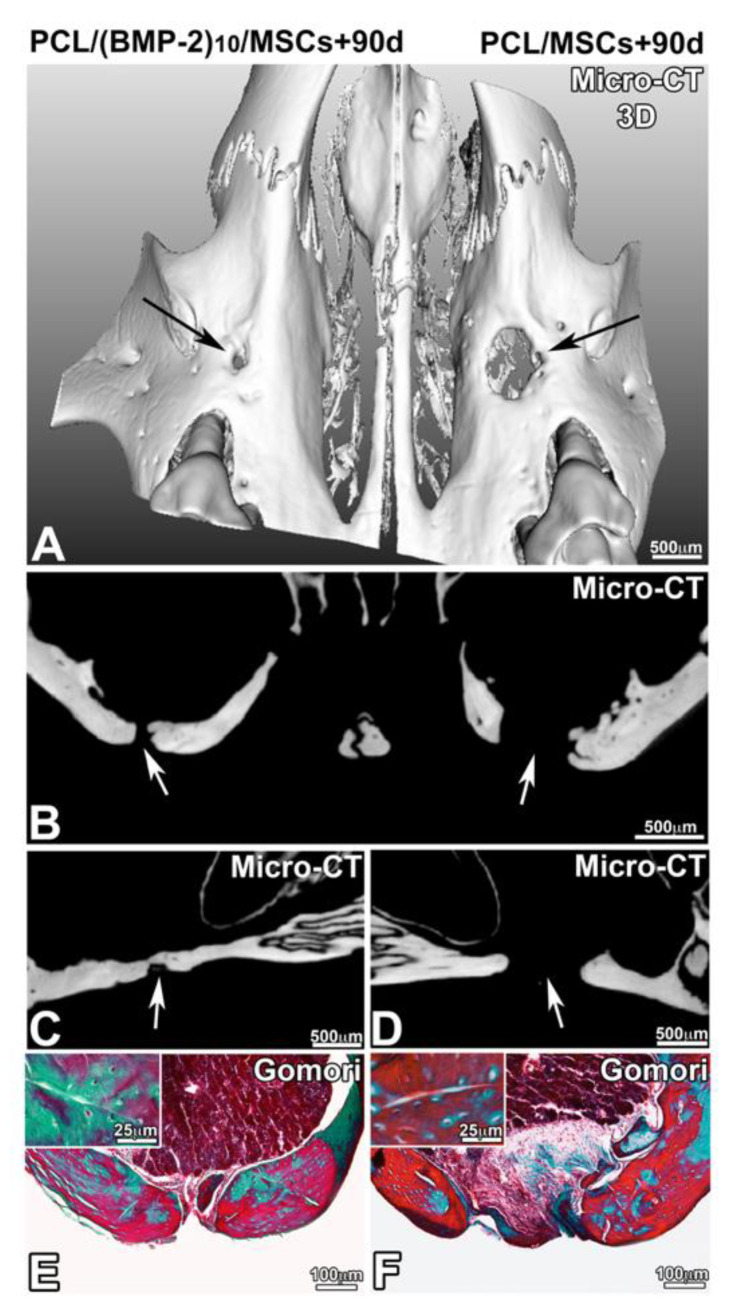
Observation of bone regeneration of the maxillary bone after 90 days of implantation of a PCL membrane functionalized with BMP-2 (**A**–**C**,**E**) and non-functionalized (**A**,**B**,**D**,**F**) with hBM-MSCs. A 3D reconstruction (**A**) of frontal (**B**) and sagittal (**C**,**D**) sections of X-ray microtomography and Gomori trichrome staining (**E**,**F**).

**Figure 6 nanomaterials-10-01774-f006:**
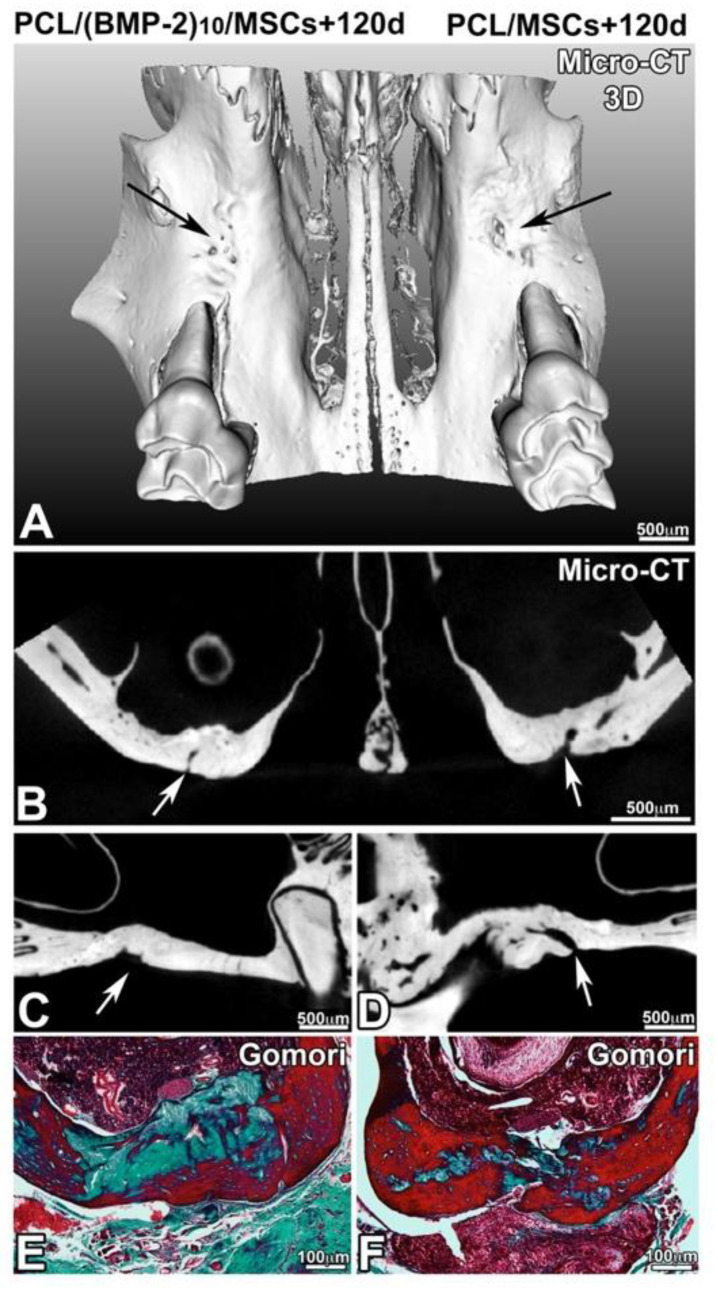
Observation of bone regeneration of the maxillary bone after 120 days of implantation of a PCL membrane functionalized with BMP-2 (**A**–**C**,**E**) and non-functionalized (**A**,**B**,**D**,**F**) with hBM-MSCs. A 3D reconstruction (**A**) of frontal (**B**) and sagittal (**C**,**D**) sections of X-ray microtomography and Gomori trichrome staining (**E**,**F**).

**Figure 7 nanomaterials-10-01774-f007:**
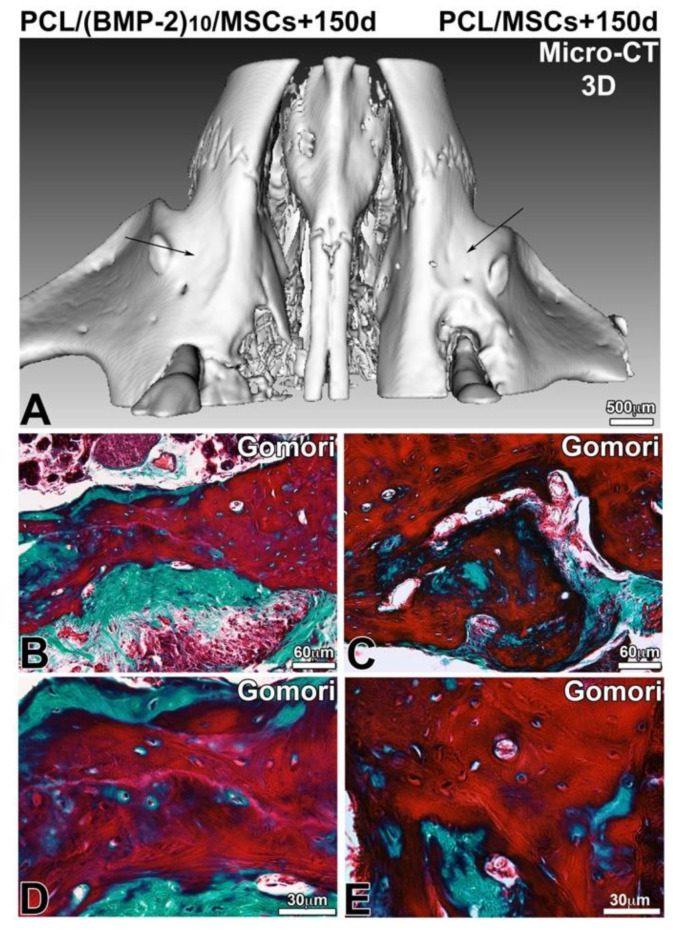
Observation of bone regeneration of the maxillary bone 150 days after implantation of a PCL membrane functionalized with BMP-2 (**A**,**B**,**D**) and non-functionalized (**A**,**C**,**E**) with hBM-MSCs. A 3D reconstruction (**A**) of frontal and sagittal sections of X-ray microtomography and after microtome cutting and Gomori trichrome staining (**B**–**E**).

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
