# Peer review of "A New Polycaprolactone-Based Biomembrane Functionalized with BMP-2 and Stem Cells Improves Maxillary Bone Regeneration"

_nanomaterials, 2020, doi:10.3390/nano10091774_

Round 1
Reviewer 1 Report
The authors of the present study excellently managed to fabricate polycaprolactone (PCL) based biomembranes by using an electrospinning approach. In the following, the membranes were functionalized with chitosan/BMP-2 nanoreservoirs. Afterwards, the membranes were supplemented with hBM-MSCs and after seven days of sub-culturing implanted in 500 um lesions placed in the maxillary bone of nude mice.
The membranes were morphologically analyzed by scanning electron microscopy. Furthermore, the cytotoxicity of the membranes was assessed. Also, the mesenchymal stem cells used in the this present study were analyzed in various ways. A colony forming test was used to show the self-renewal capacity of the cells. The multipotency of the cells was shown by differentiating the cells to osteocytes, adipocytes and chondrocytes, respectively. After the differentiating procedure, the different cells were analyzed by respective stainings.
Finally, the mice used in the in vivo experiments were sacrificed and the maxillaries analyzed histologically as well as by x-ray microCT.
The introduction illustrates the background of the study in a pleasing way. Current literature is included and the aim of the study as well as the work for the present manuscript is clearly stated. The preceding work is mentioned and clearly differentiated.
The subsequent materials and methods chapter is comprehensive and described in a good way. However, I would like to ask the authors to harmonize the supplier informations for the used materials and devices throughout the manuscript. These should always include supplier, city and country. In case of the formation of nanoreservoirs (2.2): are the chitosan and the BMP-2 also diluted in 40 mM MES pH 5.5? If not which buffer was used for dilution? Maybe, this section could be described a little bit clearer.
What was the source/supplier of the hBM-MSCs?
In the results and the adjacent discussion section, the authors show and discuss their results with the help of very appealing and valuable figures. The conclusions drawn by the authors are supported by the shown results.
In general, the presented work is of high interest and done in a very nice way. Thus, the manuscript is absolutely suitable for publication in nanomaterials.
Author Response
First, we thank the reviewer for his/her positive opinion.
The subsequent materials and methods chapter is comprehensive and described in a good way. However, I would like to ask the authors to harmonize the supplier informations for the used materials and devices throughout the manuscript. These should always include supplier, city and country. In case of the formation of nanoreservoirs (2.2): are the chitosan and the BMP-2 also diluted in 40 mM MES pH 5.5? If not which buffer was used for dilution? Maybe, this section could be described a little bit clearer.
Answer:
Changes in the text have been highlighted in yellow. In the text the name of the suppliers as well as the city and the country have been added.
In case of the formation of nanoreservoirs, the chitosan and BMP-2 are diluted in 40 mM MES pH 5.5. This was added in paragraph 2.2 and this section was described clearer. Figure 1 shows the electrospinning technique, the nanostructure of the electrospun PCL fibers using scanning electron microscopy (SEM) and the nanoreservoir functionalization technique of PCL fibers with BMP-2 (PCL/(chitosan/BMP-2)10).
What was the source/supplier of the hBM-MSCs?
Answer:
Human bone marrow derived mesenchymal stem cells (hBM-MSCs) were purchased from PromoCell, Heidelberg, Germany.
Best regards
Dr Sabine Kuchler-Bopp
Reviewer 2 Report
The research topic is very interesting and valuable for medicine and dentistry. The paper is excellently written and I suggest that it be published.
Author Response
We thank the reviewer for his/her positive opinion.
Best regards
Dr Sabine Kuchler-Bopp
Reviewer 3 Report
In the manuscript by Céline Stutz et al., authors investigate the use of BMP-2 functionalized PCL biomembranes with human bone marrow derived mesenchymal stem cells (hBM-MSCs) to regeneration of maxillary bone lesions. Indeed, the three-dimensional polycaprolactone scaffolds have a long biodegradation period and has the potential to be colonized by cells, therefore, it can be completely replaced with full-fledged bone tissue and thus will play a key role in the regeneration of major bone defects. In its turn, mesenchymal stromal cells (MSCs) have become an attractive tool for regenerative medicine due to their self-renewal, multilineage and immunosuppressive potencies as well as the ease of their isolation by standard methods. Their capability to migrate and repair injured tissues and organs makes them as a very promising tool for transplantation.
This work is relevant and has substantial interest from fundamental and practical of medical point of view.
Nevertheless, some minor modifications are required.
The authors does not specify the source of the cells - the authors themselves obtained the culture or used a commercial cell line. It is also important to note at which passage the cells were used, since this is a very important indicator for the functionality of the cells.
An analysis should be made comparing the efficacy of protein-modified PCL versus untreated PCL. Authors link to their article, but comparison analysis should be included in the text.
Also, there is also no comparison of the results of using only cells versus cells seeded on a scaffold. It would be interesting to compare them at least with the literature data.
Author Response
First, we thank the reviewer for his/her positive opinion.
The authors does not specify the source of the cells - the authors themselves obtained the culture or used a commercial cell line. It is also important to note at which passage the cells were used, since this is a very important indicator for the functionality of the cells.
Answer:
A commercial cell line was used: hBM-MSCs were purchased from PromoCell, Heidelberg, Germany and used from passage 4 to 6. This was added in paragraph 2.3.
An analysis should be made comparing the efficacy of protein-modified PCL versus untreated PCL. Authors link to their article, but comparison analysis should be included in the text.
Answer:
The comparison of the efficacy of PCL-BMP-2 versus untreated PCL has been added in the text in the results section 3.3. In the results section we added: “After 120 days, the efficacy of PCL-BMP-2 versus untreated PCL efficacy on bone remodeling has been also tested. Micro-CT and histology results have shown that the presence of BMP-2 accelerates the regeneration of maxillary bone (Supplementary Figure 1).”
Also, there is also no comparison of the results of using only cells versus cells seeded on a scaffold. It would be interesting to compare them at least with the literature data.
Answer:
In the field of tissue regeneration such as bone and cartilage, scaffold provides a support for the regeneration of new tissue regardless of the benefit of stem cells, which is why we have chosen to use a scaffold. On the other hand, the culture of stem cells on the scaffold allows precise control of the administration of stem cells at the defect level compared to an injection. This is why we chose to use a scaffold rather than injections using for example osteogenic cell sheets fabricated from BMSCs (Shimizu et al., Injury 2015, 1457-1464) or hydrogels seeded with BMSCs (Zhang et al., J Mater Chem B 2018 6(13):1951-1964).
Best regards
Dr Sabine Kuchler-Bopp